# Four Types of Multiclass Frameworks for Pneumonia Classification and Its Validation in X-ray Scans Using Seven Types of Deep Learning Artificial Intelligence Models

**DOI:** 10.3390/diagnostics12030652

**Published:** 2022-03-07

**Authors:** Pankaj K. Jain, Neeraj Sharma, Mannudeep K. Kalra, Klaudija Viskovic, Luca Saba, Jasjit S. Suri

**Affiliations:** 1School of Biomedical Engineering, Indian Institute of Technology (BHU), Varanasi 221005, India; nillmani.rs.bme17@itbhu.ac.in (N.); pankajkrjain.rs.bme17@itbhu.ac.in (P.K.J.); neeraj.bme@itbhu.ac.in (N.S.); 2Department of Radiology, Massachusetts General Hospital, Boston, MA 02115, USA; mkalra@mgh.harvard.edu; 3Department of Radiology and Ultrasound, University Hospital for Infectious Diseases, 10000 Zagreb, Croatia; klaudija.viskovic@bfm.hr; 4Department of Radiology, Azienda Ospedaliero Universitaria (A.O.U.), 10015 Cagliari, Italy; lucasabamd@gmail.com; 5Stroke Diagnostic and Monitoring Division, AtheroPoint^TM^, Roseville, CA 95661, USA; 6Knowledge Engineering Center, Global Biomedical Technologies, Inc., Roseville, CA 95661, USA

**Keywords:** COVID-19, Omicron, chest X-rays, deep learning, transfer learning, convolutional neural network

## Abstract

Background and Motivation: The novel coronavirus causing COVID-19 is exceptionally contagious, highly mutative, decimating human health and life, as well as the global economy, by consistent evolution of new pernicious variants and outbreaks. The reverse transcriptase polymerase chain reaction currently used for diagnosis has major limitations. Furthermore, the multiclass lung classification X-ray systems having viral, bacterial, and tubercular classes—including COVID-19—are not reliable. Thus, there is a need for a robust, fast, cost-effective, and easily available diagnostic method. Method: Artificial intelligence (AI) has been shown to revolutionize all walks of life, particularly medical imaging. This study proposes a deep learning AI-based automatic multiclass detection and classification of pneumonia from chest X-ray images that are readily available and highly cost-effective. The study has designed and applied seven highly efficient pre-trained convolutional neural networks—namely, VGG16, VGG19, DenseNet201, Xception, InceptionV3, NasnetMobile, and ResNet152—for classification of up to five classes of pneumonia. Results: The database consisted of 18,603 scans with two, three, and five classes. The best results were using DenseNet201, VGG16, and VGG16, respectively having accuracies of 99.84%, 96.7%, 92.67%; sensitivity of 99.84%, 96.63%, 92.70%; specificity of 99.84, 96.63%, 92.41%; and AUC of 1.0, 0.97, 0.92 (*p* < 0.0001 for all), respectively. Our system outperformed existing methods by 1.2% for the five-class model. The online system takes <1 s while demonstrating reliability and stability. Conclusions: Deep learning AI is a powerful paradigm for multiclass pneumonia classification.

## 1. Introduction

COVID-19 is an extremely contagious disease caused by severe acute respiratory syndrome coronavirus 2 (SARS-CoV-2) [1]. The virus was first isolated from three pneumonia patients with critical respiratory illness in December 2019 in Wuhan, China [2]. Within a short period, the virus spread globally. On 11 March 2020, World Health Organization (WHO) declared the disease a pandemic [3]. Coronaviruses (CoVs) are a tremendously diverse family of enveloped positive-sense single-stranded RNA viruses [4]. The viruses are highly pathogenic and transmissible viruses that spread via respiratory droplets or aerosol between individuals in close proximity [5], leading to several pathways [6] causing damage to several organs such as heart [7] and liver [8], causing diabetes [9] and pulmonary embolism [10,11]. In the majority of infected cases, the person begins to exhibit symptoms such as cough, fever, fatigue, and loss of smell or taste. In numerous fatal instances, the infection progresses to the lower respiratory system, including the lungs, causing illness such as severe pneumonia followed by multi-organ dysfunction syndrome with several secondary infections and shock [12,13,14,15,16,17].

Even after two years of the virus outbreak and almost 10,000 million doses of vaccination being administered, the disease continues to destroy human health, life, and the global economy. The viruses are incredibly efficient in mutating fast and gradually converting into more deadly variants [18]. After the severe damage of the Delta variant, a new variant named Omicron was discovered. The WHO has already designated Omicron as a variant of concern [19]. Several notable mutations in spike proteins of Omicron make it highly transmissible. Moreover, there is still a risk of more new mutations in Cov-2 thereafter, presenting potential for a more pernicious variant outbreaks.

COVID-19 infection is normally detected by a reverse transcriptase polymerase chain reaction (RT-PCR) test, which is frequently followed by chest radiographs, such as X-rays and computed tomography (CT) scans [20,21]. The reference technique for COVID-19 detection is RT-PCR; although, the procedure is laborious, complicated, rigorous, and time consuming with a significant high error rate [20,22,23]. The RT-PCR kit, along with a specific biosafety facility to host the PCR machine, is expensive. Consequently, there is a substantial supply constraint. Many nations are experiencing problems with erroneously positive COVID-19 cases caused by inadequacy in test kit supply, as well as delays in the test results. These limitations of RT-PCR present major obstacles to restricting the control of the disease as infection spreads among healthy populations [24].

To counteract the spread of COVID-19, patients must undergo prompt and effective screening, as well as get appropriate medical attention. Several medical imaging modalities, including chest X-ray (CXR) and computed tomography (CT), can help with this [25,26]. COVID-19 has recently been detected using CT imaging [25,27], however, the high patient dosage and screening expenses are principal disadvantages of using CT imaging for diagnosis [28]. On the other hand, the CXR equipment is commonly accessible in hospitals and diagnostic centers to create a 2D projection of the thorax quickly and affordably. Radiologists already use the CXR modality to detect chest abnormalities in various lung illnesses, including pneumonia and tuberculosis. COVID-19 detection has also been done utilizing CXRs in a few patients [25,29]. COVID-19 patients reveal similar findings in radiographs such as bilateral, peripheral, and basal predominant ground-glass opacities, septal thickening, pleural effusion, bronchiectasis, and bilateral lymphadenopathy [27,30,31,32,33,34,35]. As a result, CXR scans might help in the early detection of COVID-19 in the suspected person. However, one challenge is that the CXRs of various pneumonia are very similar; therefore, it is tough to differentiate COVID-19 from other lung abnormalities manually. Nonetheless, deep learning algorithms powered by Artificial Intelligence (AI) can efficiently extract several image-based features that radiologists may be unable to observe manually in the original CXR. Regarding image feature extraction and classification, convolutional neural networks (CNNs) have proven their efficiency and are widely implemented by the research community [36,37,38,39,40,41,42,43,44,45,46,47,48,49,50,51,52,53,54,55,56]. Nowadays, CNN-based solutions are widely utilized to resolve a variety of health problems such as brain tumor identification [57,58,59], lung and breast cancer detection [60,61,62], Alzheimer’s disease diagnosis [63], cardiovascular disease predictions [64,65,66,67,68,69,70], pneumonia detection [71,72,73,74,75], and many more. With the promising results in several applications, deep learning techniques for chest X-rays have been gaining prominence in recent years. The transfer learning technique has made the operation smoother by facilitating the quick retraining of a highly deep CNN [76,77,78,79,80,81,82,83,84,85,86,87].

In this work, we have designed and applied seven different deep learning models utilizing the transfer learning method to detect multiclass COVID-19 in CXR images. We have performed the binary and multiclass classification into COVID-19 and other lung diseases—namely, viral pneumonia (VP), bacterial pneumonia (BP), tuberculosis (TB), and normal images. Thereafter, we compared the results to get the best-suited model for their usefulness in practice. Figure 1 shows the overall schematic diagram of the development of the COVID-19 detection system.

The whole work has been structured in a section-wise manner. In Section 2, we have explored all the related work and contributions of different authors in this area. Section 3 explains the dataset, image-preprocessing, and deep learning models. In Section 4, results of the experiments and their comparative performances have been provided. Section 5 deals with the model’s performance evaluation. Next, Section 6 presents the scientific validation of the proposed models which has been done on another dataset. Further, in Section 7, we have compared the proposed models with other existing state-of-the-art methods. Finally, Section 8 concludes the study and presents the future scope.

## 2. Related Work

Recently, COVID-19 detection using deep learning techniques has become a very popular area of research. Several researchers have proposed deep learning methods for the detection of disease in CXR images. However, the majority of them employed a limited dataset with a small number of COVID-19 samples. Consequently, their outputs may not be generalized, and accuracy cannot be guaranteed with a larger dataset. Choudhury et al. [88] applied eight different deep learning pre-trained CNNs for the classification of CXR images with three classes named COVID-19, viral pneumonia, and normal, with a total of 423, 1485, and 1579 images for each class, respectively. The authors showed an accuracy of 97.74% by CheXNet for three-class with the equivalent precision, sensitivity, and F1-score of 96.61%, and specificity of 98.31%. Hemdan et al. [89] utilized 50 CXR images with 25 confirmed COVID-19 and 25 normal for classification using pre-trained deep CNNs and achieved a maximum accuracy of 90% using VGG16 and DenseNet201 models with the precision of 83%, recall of 100%, and F1-score of 91% for both of the networks. Hussain et al. [90] developed a novel deep neural network (DNN) named CoroDet. The authors used CXR images under four classes named COVID-19, viral pneumonia (VP), bacterial pneumonia (BP), and normal with an sample sizes of 500, 400, 400, and 800 for each class respectively. They performed the classification experiment into two-class (COVID-19 vs. normal), three-class (COVID-19, VP, and normal), and four-class (COVID-19, VP, BP, and normal) models with maximum accuracies of 99.1%, 94.2%, and 91.2% for each experiment respectively. Jain et al. [91] applied several pre-trained CNNs for the classification of CXR images into three classes—COVID-19, VP, and normal. They utilized 490 COVID-19 images and got the maximum accuracy of 97.97% using the Xception model. Mahdy et al. [92] recommended a deep CNN-based methodology for COVID-19 detection from chest X-ray images with an accuracy of 97.48%. Ioannis et al. [93] applied transfer-learning methods to classify CXR images into COVID-19, BP, and normal classes with 224, 700, and 504 images for each class, respectively. They attained 96.7% accuracy, 98.66% sensitivity, and 96.46% specificity for the experiment. Sethy et al. [94] applied ResNet50 and SVM to classify CXRs into COVID-19, pneumonia, and normal classes. They obtained an accuracy of 95.33% for the three-class experiment. Ozturk et al. [95] introduced a novel network named DarkCovidNet. Using this network, the authors received an accuracy of 98.08% for two-class classification and 87.02% accuracy for three-class classification. Khan et al. [96] introduced a novel network: Coronet inspired from Xception architecture. Using the Coronet model, the authors obtained an accuracy of 95% for three-class classification into COVID-19, VP, and normal. They also performed four-class classification into COVID-19, VP, BP, and normal with 89.6% accuracy. Wang et al. [97] introduced a novel DNN, named COVID-Net, for the detection of COVID-19. The authors utilized 13,975 CXR images for the classification and achieved an accuracy of 83.5%. Afshar et al. [98] introduced COVID-CAPS, a capsule network to classify small-sized data of CXR images. The authors obtained an accuracy of 95.7% using COVID-CAPS. Yang et al. [99] applied transfer four different learning-based networks to classify CXR images into binary and three-class. The authors obtained an accuracy of 99% for binary (COVID-19 and pneumonia) and 97% accuracy for three-class (COVID-19, pneumonia, and normal) classification, both by the VGG16 network. Nayak et al. [100] performed binary classification into COVID-19 and normal class using 406 CXR images. The authors applied eight different pre-trained neural networks using the transfer learning method and obtained a maximum accuracy of 98.33% using ResNet34 network. When it comes to fusion of machine learning and deep learning, Bhattacharya et al. [101] performed three-class classification. This was aimed at classification of CXRs into COVID-19, pneumonia, and normal class. The authors obtained a maximum accuracy of 96.6% using a combination of VGG16 and binary robust invariant scalable key-points algorithm. Deb et al. [102] proposed a multi-model deep CNN ensemble architecture for the classification of CXRs into binary (COVID-19 and non-COVID-19) and three-class (COVID-19, pneumonia, and normal). The authors obtained accuracies of 98.58% for binary and 93.48% for the three-class experiment. Nikolaou et al. [103] developed a novel CNN by modifying pre-trained EfficientNetB0. This network was applied for the binary (COVID-19 and normal) and three-class (COVID-19, pneumonia, and normal) classification, obtaining an accuracy of 95% for binary and 93% for three-class. Oh et al. [104] introduced a patch-based DNN, where the network was applied for four-class classification of CXRs into COVID-19, BP, TB, and normal. Their database consisted of 502 images, where 180 were COVID-19 images, and they obtained a classification accuracy of 88.9%. AI-Timemy et al. [105] performed five-class classification divided into COVID-19, VP, BP, TB, and normal class. They utilized 2186 images, including 435 COVID-19 images, for the experiment. The authors applied a combination of DL and ML methods and attained 91.6% accuracy.

In conclusion, several recent studies have been reported for COVID-19 and other pneumonia classifications using CXR images. Most of them applied various CNN networks and achieved promising results. However, in maximum cases, the dataset used has a deficient number of images due to the scarcity of COVID-19 data. Hence, their results need to be verified on a larger dataset. Additionally, the classification into relevant multiclasses (>3 pneumonia) is rare. A rigorous experiment on classification for a larger dataset of COVID-19 and other similar lung disorders is required. In this study, we have designed and applied seven different deep learning models utilizing the transfer learning method for the classification of four types of pneumonia including COVID-19. We have used almost the largest data set of 18,603 CXR images which consists of 3611 COVID-19, 1345 viral pneumonia, 2780 bacterial pneumonia, 700 tuberculosis, and 10,167 normal CXR images.

## 3. Methodology

We have designed and applied seven highly efficient pre-trained deep CNNs for the binary and multiclass classification of pneumonia diseases. The approaches we have opted for in this experiment are described in the five subsequent sub-sections.

### 3.1. Dataset

In this experiment, 18,603 CXR images were used, including both anterior-to-posterior (AP)/posterior-to-anterior (PA). The dataset was prepared from three different publicly available databases. COVID-19, viral pneumonia, and normal CXR images were taken from the Kaggle: “COVID-19 Radiography Database”, i.e., winner of the COVID-19 Dataset Award by Kaggle Community [106]. The tuberculosis images were taken from the Kaggle: “Tuberculosis (TB) Chest X-ray Database” [107]. Finally, the bacterial pneumonia images were taken from the Kaggle: “Chest X-Ray Images (Pneumonia)” [108].

#### 3.1.1. COVID-19 Radiography Database

The COVID-19 radiography database includes CXR images of COVID-19 and viral pneumonia patients along with healthy persons. The dataset was created by different research groups and doctors in collaboration [88,109]. The first stage of release of the dataset had 219 COVID-19, 1341 normal, and 1345 viral pneumonia chest X-rays. After two updates, the current dataset has increased the number up to 3616 COVID-19, 10,192 normal, and 1345 viral pneumonia images. The images were in Portable Network Graphics (PNG) file format with a resolution of 299 × 299 pixels. We have taken all the COVID-19, viral pneumonia, and normal images for our experiment.

#### 3.1.2. Chest X-ray Pneumonia Images

The chest X-ray images (pneumonia) dataset contains 5863 CXR images with 2780 bacterial pneumonia and the rest with viral pneumonia and normal images. The CXR images were taken from Guangzhou Women and Children’s Medical Center, Guangzhou, China [110,111]. The images were in JPEG format with variable resolutions. We have taken all the 2780 bacterial pneumonia images for our experiment.

#### 3.1.3. Tuberculosis Chest X-ray Database

The tuberculosis chest X-ray database contained CXR images of tuberculosis patients along with the healthy person. The dataset was created by several research groups along with the collaboration of medical doctors [112]. There are 700 tuberculosis images in Portable Network Graphics (PNG) file format with a resolution of 512 × 512 pixels. We have taken all the 700 tuberculosis images for our experiment. Figure 2 shows sample CXR images from each class. The images indicate that it is hard to determine the differences between them manually.

### 3.2. Image Processing

All the CXR images collected from the different data sources were first converted into Portable Network Graphics (PNG) file format. Out of 18,632, 29 images, i.e., <1%, were excluded from the experiment as outliers since they were missing details such as lung region. Some X-ray images containing avoidable body parts were cropped, displaying only chest and lungs. Image augmentation was done for each image included in the training process. During image augmentation, shearing and zooming were applied to 20%, typically adapted in the imaging industry [113,114,115,116]. Images were resized to 224 × 224 pixels before the training process as required for the pre-trained model standards.

Finally, a total of 18,603 CXR images including 3611 COVID-19, 1345 viral pneumonia, 2780 bacterial pneumonia, 700 tuberculosis, and 10,167 normal images were utilized for the experiments. Table 1 shows the experimental steps and class-wise distribution of the images. Out of the total, 80% (i.e., 14,879 images) including 2887 COVID-19, 1075 viral pneumonia, 2224 bacterial pneumonia, 560 tuberculosis, and 8133 normal images were utilized for training the models. Next, 10%, (i.e., 1862) randomly selected images including 362 COVID-19, 135 viral pneumonia, 278 bacterial pneumonia, 70 tuberculosis, and 1017 normal images were utilized for validation. Finally, 10%, (i.e., 1862) randomly selected images that were not involved in training or validation were utilized to test the models. The test set included 362 COVID-19, 135 viral pneumonia, 278 bacterial pneumonia, 70 tuberculosis, and 1017 normal images.

### 3.3. Experimental Setup

The whole experiment was organized into three phases. During the first phase of the experiment, we classified the images into two classes as: (i) COVID-19 and normal, (ii) COVID-19 and viral pneumonia, (iii) COVID-19 and bacterial pneumonia, and (iv) COVID-19 and tuberculosis. In the second phase of the experiment, we performed the three-class classification into viral diseases i.e., COVID-19, viral pneumonia and normal. In the third and final phase of the experiment, five-class classification was done into viral and bacterial diseases—i.e., COVID-19, viral pneumonia, bacterial pneumonia, tuberculosis, and normal. The experimental protocol consisted of 80% training, 10% validation, and 10% testing. The experiment was performed utilizing Python 3.8 on a computer with Intel Core i7 8th Generation Processor, 16 GB RAM, and 8 GB NVIDIA Quadro P4000 graphics processing unit (GPU).

### 3.4. Model Architectures

Transfer learning is a machine learning approach in which a model developed for one job is used as the foundation for another task. It uses a trained model from a large dataset. Pre-trained weights are then used to train the network more quickly for an application with a smaller dataset. This eliminates the need for a large dataset and shortens the training time that a deep learning system requires when created from scratch. In this work, utilizing the transfer learning approach we applied seven highly efficient pre-trained CNNs—namely, VGG16, VGG19, Xception, InceptionV3, Densenet201, NasnetMobile, and Resnet152—for the experiment. The architecture of each network is shown in Figure 3a–g. The Densenet offers a superior architecture design when it comes to layering process. The feature maps of the preceding layers are utilized in all the subsequent layers. This reduces the complexity drastically, thereby improving the performance. In a conventional network, there are M connections for M layers, unlike in the dense layer there are M(M + 1)/2 direct connections, hence it is powerful and efficient. The loss function applied for two-class was binary cross-entropy and for multiclass was categorical cross-entropy (CE). The activation function applied for the dense layer was sigmoid for binary and softmax for multiclass classification. The output layer was modified according to the number of classes. The models were trained for 25 epochs with a batch size of 16 images.

### 3.5. Cross-Entropy Loss Function for Models

Binary cross-entropy loss function can be defined as in Equation (1) [67].
(1)$$L_{BCE}=\frac{-1}{N}\sum_{i=1}^{N}\left[\left(y_{i}\times \log a_{i}\right)+\left(1-y_{i}\right)\times \log(1-a_{i}\right)]$$
where, *y_i_* is the input GT label 1, (1 − *y_i_*) is GT label 0, $a_{i}$ represents the Softmax classifier probability.

Categorical cross-entropy loss function can be defined as in Equation (2) [67].
(2)$$L_{CCE}=\frac{1}{N}\sum_{i=1}^{N}\sum_{c=1}^{C}1_{y_{i}\in C_{c}}\log a_{model}(y_{i}\in C_{c})$$
where, *N* is the total number of observations (images), *C* is the number of categories or classes, $1_{y_{i}\in C_{c}}$ term indicates the *i*^th^ observation that belongs to the *c*^th^ category.

### 3.6. Performance Metrics Used for Classification Evaluation

The performances of the proposed models were evaluated by the following different matrices:(a)Accuracy: Accuracy is the most significant criterion for the analysis of the convolutional neural network’s performance. Accuracy is the sum of true positive and true negative values divided by the entire component of the confusion matrix. It is represented as given in Equation (3) [88].
(3)$$\mathrm{Accuracy}=\frac{\mathrm{True}\;\mathrm{Positives}+\mathrm{True}\;\mathrm{Negatives}}{\mathrm{Total}\;\mathrm{number}\;\mathrm{of}\;\mathrm{cases}}$$

(b)Precision: Precision is an important measure of the results of the CNN models. It counts how many correct positive predictions have been made. Precision is evaluated as the ratio between true positive predicted components and the sum of positive predicted components. It is represented as given in Equation (4) [88].


(4)
$$\mathrm{Precision}=\frac{\mathrm{True}\;\mathrm{Positive}}{\mathrm{True}\;\mathrm{Positive}+\mathrm{False}\;\mathrm{Positive}}$$


(c)Recall (Sensitivity): Recall is another important metric for the analysis of the classifier’s performance. It is defined as the ratio between the true positive predicted components and the sum of true positive and false negative predicted components. It is represented as given in Equation (5) [91].


(5)
$$\mathrm{Recall}=\frac{\mathrm{True}\;\mathrm{Positive}}{\mathrm{True}\;\mathrm{Positive}+\mathrm{False}\;\mathrm{Negative}}$$


(d)F1-score: The F1-score is an important measure for assessing the test’s accuracy. It is the harmonic mean between precision and recall. It is defined as twice the ratio between multiplication of precision and recall to the sum of precision and recall. It is represented as given in Equation (6) [91].


(6)
$$F1-\mathrm{score}=\frac{\;2\;\times \;\mathrm{Precision}\;\times \;\mathrm{Recall}}{\mathrm{Precision}+\mathrm{Recall}}$$


## 4. Results

Three different phases of the experiment were performed to compare the results of each classification possibility. In the first phase, we performed binary, then three-class, and finally five-class classification experiments.

### 4.1. Binary Classification

The binary classification experiment deals with the classification of images into COVID-19 and other classes separately. We endeavored to know how accurately the models could classify the images of different classes from the COVID-19 class. The binary experiment was stepped into four different sub-phases as COVID-19 vs. normal, COVID-19 vs. viral pneumonia, COVID-19 vs. bacterial pneumonia, and COVID-19 vs. tuberculosis classification.

#### 4.1.1. Binary Class Case 1: COVID-19 vs. Normal

The comparative performances of different CNNs for the binary classification into COVID-19 and normal images are shown in Table 2. Using Equations (3)–(6), we evaluated the performance of the VGG 16 network, and it demonstrated the greatest efficiency with the highest accuracy, precision, recall, and f1-score among all networks. The VGG16 achieved a test accuracy of 97.24% with weighted averages of precision, recall, and f1-score of 97.26%, 97.24%, and 97.21%, respectively. The DenseNet201 performed as the second most efficient network with an accuracy of 96.01%. The performance of ResNet152 was least efficient with an accuracy of 78.75%. Figure 4 shows the training and validation accuracies and Figure 5 shows the training and validation loss for the best performing VGG16 model. The graphs indicate improved accuracy and reduced loss with successive epochs. Figure 6 shows the confusion matrix of test data classification by the VGG16 model. The confusion matrix specifies that, out of 362 COVID-19 images, 331 were correctly classified, and 31 were misclassified as normal images. Whereas, out of 1017 normal images, 1010 were correctly predicted, and seven were misclassified as the COVID-19 images.

#### 4.1.2. Binary Class Case 2: COVID-19 vs. Viral Pneumonia

Table 3 shows the comparative performances of different CNNs for binary classification into COVID-19 and viral pneumonia. The NasnetMobile network performed most efficiently with the highest accuracy, precision, recall, and f1-score among all networks. The model achieved an accuracy of 99.80% with the equivalent weighted average of precision, recall, and f1-score of 99.80% each. VGG16 model performed as the second most efficient network with an accuracy of 99.60%. The performance of the ResNet152 model was least efficient, with an accuracy of 97.79%.

Figure 7 shows the training and validation accuracy and Figure 8 shows the training and validation loss for the best performing NasnetMobile model. The graphs show how accuracy improves and loss reduces with successive epochs. Figure 9 shows the confusion matrix of the test data classification by the NasnetMobile model. The confusion matrix reveals that, out of 362 COVID-19 images, 361 were correctly predicted, and one was misclassified to the viral pneumonia class. Furthermore, our model correctly predicted all 135 viral pneumonia images without any false prediction.

#### 4.1.3. Binary Class Case 3: COVID-19 vs. Bacterial Pneumonia

The comparative performance metrics of different CNNs for binary classification into COVID-19 and bacterial pneumonia are shown in Table 4. The DenseNet201 performed most efficiently with the highest accuracy, precision, recall, and f1-score among all networks. The model achieved an accuracy of 99.84% and an equivalent weighted average of precision, recall, and f1-score of 99.84% each. The InceptionV3 and NasnetMobile performed as the second most efficient network with the equivalent accuracy of 99.53%. The ResNet152 performed least efficiently with an accuracy of 98.59%.

Figure 10 shows the training and validation accuracy, and Figure 11 shows the training and validation loss for the best performing DenseNet201 model. The graphs show that accuracy improves and loss reduces with successive epochs.

Figure 12 shows the confusion matrix of the test data classification by the DenseNet201 model. The confusion matrix specifies that out of 362 COVID-19 images, 361 were correctly predicted and one image was misclassified to bacterial pneumonia class. However, the model correctly predicted all 278 bacterial pneumonia images without any false predictions.

#### 4.1.4. Binary Class Case 4: COVID-19 and Tuberculosis

The comparative performance metrics of different CNNs for binary classification into COVID-19 and tuberculosis CXR images have shown in Table 5. VGG16 performed most efficiently with an accuracy of 99.31%, weighted average of precision and recall of 99.31%, and f1-score of 99.30%. VGG19 and Xception both performed as the second most efficient models with the equivalent accuracy of 99.07%. ResNet152 performed least efficiently with an accuracy of 91.20%.

Figure 13 shows the training and validation accuracy and Figure 14 shows the training and validation loss for the best performing VGG16 model. The graphs indicate improved accuracy and reduced loss with successive epochs. Figure 15 shows the confusion matrix of the test data classification by the VGG16 model. The confusion matrix reveals the model correctly classified all 362 COVID-19 CXR images. Furthermore, out of 70 tuberculosis CXR images, 67 were correctly predicted, and three were misclassified as COVID-19 images.

### 4.2. Three-Class Classification into Viral Diseases

The comparative performance metrics of different CNNs for three-class classification into COVID-19, viral pneumonia, and normal images have shown in Table 6. VGG16 network performed most efficiently with an accuracy of 96.63% and equivalent weighted average of precision, recall, and f1-score of 96.63% each. The DenseNet201 network performed as the second most efficient network with an accuracy of 95.51%. The performance of ResNet152 model was the least efficient with an accuracy of 77.21%.

Figure 16 shows the training and validation accuracy and Figure 17 shows the training and validation loss for the best performing VGG16 model. The graphs indicate improved accuracy and reduced loss with successive epochs.

Figure 18 shows the confusion matrix of the test data classification by the VGG16 model. The confusion matrix specifies that out of 362 COVID-19 images, 339 were correctly classified, and 23 were misclassified as 21 to normal and two images to the viral pneumonia class. Next, out of 1017 normal images, 994 were correctly predicted, and 23 were misclassified with 18 as COVID-19 and five images as viral pneumonia class. Further, out of 135 viral pneumonia images, 130 were correctly classified, and five were misclassified as normal images.

### 4.3. Five-Class Classification into Viral and Bacterial Diseases

The comparative performance metrics of different networks for classification into five classes: COVID-19, viral pneumonia, bacterial pneumonia, tuberculosis, and normal images are shown in Table 7. The VGG16 model performed most efficiently with an accuracy of 92.70% and weighted averages of precision, recall, and f1-score of 92.41%, 92.70%, and 92.47%, respectively. The DenseNet201 performed as the second most efficient model with an accuracy of 89.10%. The performance of ResNet152 network was least efficient, with an accuracy of 74.70%.

Figure 19 shows the training and validation accuracy and Figure 20 shows the training and validation loss for the best performing VGG16 model. The graphs indicate that accuracy improves and loss reduces with successive epochs. Figure 21 shows the confusion matrix of the test data classification by the VGG16 model. The confusion matrix reveals that, out of 362 COVID-19 images, 336 were correctly predicted and 26 were misclassified as 24 to normal, one to bacterial pneumonia, and one to tuberculosis class. Next, out of 278 bacterial pneumonia images, 238 were correctly classified and 40 were misclassified as 14 normal and 26 viral pneumonia images. Furthermore, out of 1017 normal, 1002 were correctly predicted and 15 were misclassified as 12 COVID-19, two as bacterial pneumonia, and one as a viral pneumonia image. Afterward, out of 70 tuberculosis images, 69 were correctly classified and one image was misclassified to normal class. Finally, out of 135 viral pneumonia images, 81 were correctly predicted, and 54 were misclassified as 37 bacterial pneumonia, 16 normal, and one COVID-19 image.

## 5. Performance Evaluation

We are able to design a multiclass system for COVID-19 classification and detection. The results of each experiment show very encouraging numbers. However, the system needs some performance evaluation to prove its robustness against all odds. Therefore, we obtained the receiver operating characteristic (ROC) curve and the area-under-the-curve (AUC) for the best performing model in all classification experiments.

The ROC curves are drawn using inference values and true labels for each class. Figure 22 shows the four ROC curves and AUC values for best performing models in two-class experiments. Figure 23 shows ROC curves and AUC values for the best performing model (VGG16) in three-class experiments. Similarly, Figure 24 shows ROC curves and AUC values for the best performing model (VGG16) in five-class experiments.

## 6. Scientific Validation

Scientific validation is a significant integrated part of system design. The goal of optimal model validation is to ensure that the model is also functioning well and delivering comparable results on different dataset domains. In this work, we verified all of our models on the facial biometric dataset named Faces95 from Libor Spacek’s Facial Images Databases [117]. Several articles in the literature demonstrate the use of a well-known and standardized Faces95 database [118]. The database contains 72 individual images with various expressions and positions sat at a fixed distance from the camera. There are 72 classes for both men and women, with a total of 1440 photographs. The sample images from the first eight classes are shown in Figure 25.

The experiments were performed under a similar experimental condition as for CXR images classification. The loss function applied was categorical cross-entropy. The activation function used for the dense layer was softmax. The models were trained for 25 epochs with a batch size of 16 images. The training, validation, and testing were done on 80%, 10%, and 10% of the randomly selected images, respectively. The performances were also evaluated in terms of accuracy, precision, recall, and F1-score. Table 8 shows the comparative performance of the models. The VGG16 model performed most efficiently with an accuracy of 98.61% and precision, recall, and F1-score of 99.07%, 98.61%, and 98.52%, respectively. Figure 26 shows the training and validation accuracy and Figure 27 shows the training and validation loss for the best performing VGG16 model. The graphs indicate improved accuracy and reduced loss with successive epochs. The results support our system performing excellently on other datasets too, along with the medical images, providing outstanding results in each scenario.

## 7. Discussion

The coronavirus pandemic and its fast spread has put the world in a very tough situation. The quality of life due to COVID-19 has gotten worse, including increased anxiety and depression [119]. The chaotic behavior of COVID-19 has caused it to spread as a nonlinear infection throughout different parts of the world [120]. Certain places of the world have a larger number of infections, while some have lower numbers. Furthermore, some have greater intensity and severity, while at some places it has smaller intensity and severity. Additionally, regarding cases related on a daily basis, studies have shown no significant correlation in the diffusion of disease in the different parts of the world [121]. Thus, it makes it difficult to predict, prepare, and combat the outbreak of the disease variants. To find solutions and overcome limitations—such as unavailability of RT-PCR tests, delayed results, and high costs—our strategy of deep learning-based transfer models for the classification of chest X-ray images to detect COVID-19 is proving to be most effective. We utilized 18,603 CXR images, with 3611 having COVID-19 and remainder of the sample being composed of patients with viral pneumonia, bacterial pneumonia, and tuberculosis disease classes along with normal images. We organized our experiment into three phases: (i) binary classification (COVID-19 and other classes separately); (ii) three-class classification into viral diseases (COVID-19, viral pneumonia, and normal); and (iii) five-class classification into viral and bacterial diseases (COVID-19, viral pneumonia, bacterial pneumonia, tuberculosis, and normal). To achieve optimal performance, we applied seven highly efficient pre-trained CNNs—VGG16, VGG19, DenseNet201, Xception, InceptionV3, NasnetMobile, and ResNet152—for the classification of the CXR images.

### 7.1. Principal Findings

For binary classification, we achieved the best performance by the DenseNet201 model with an accuracy of 99.84% for COVID-19 and bacterial pneumonia classification. Thereafter, the second-best performing model was NasnetMobile which provided 99.80% accuracy for the classification of COVID-19 and viral pneumonia. Finally, the VGG16 model performed third-best with 99.31% and 97.24% accuracies for classification into COVID-19 vs. tuberculosis and COVID-19 vs. normal class respectively. For three-class and five-class experiments, the VGG16 model performed best with accuracies of 96.63% and 92.70% respectively. The AUC for binary classification was best for COVID-19 vs. viral pneumonia and COVID-19 vs. tuberculosis class with the value of 1.0. Next, the AUC achieved for COVID-19 and tuberculosis was 0.98 and for COVID-19 and normal class was 0.95. Furthermore, for three-class and five-class classification, the AUC values achieved were 0.97 and 0.92 respectively.

### 7.2. Benchmarking

Table 9 shows the benchmarking table, presenting existing state-of-art classification methods and comparing them against the proposed method. Each row in the table shows different authors’ work in this area and the columns show the methods, number of X-ray images used, and results of the experiment. We used the highest number of images for our experiment compared with any other work in this area. To the best of our knowledge, our NasnetMobile model achieved the highest accuracy of 99.80% among all existing methods for binary classification of COVID-19 vs. viral pneumonia. Additionally, for the first time ever, we have performed the binary classification into COVID-19 vs. bacterial pneumonia and COVID-19 vs. tuberculosis disease classes with remarkable accuracies of 99.84% by Densenet201 and 99.31% by VGG16 model, respectively. Our results are very consistent with the previous studies on Densenet [122,123,124,125]. These studies have shown superior performance of Densenet169 applied to COVID CT/X-rays. The key advantages of the Densenet are its ability to alleviate the fundamental problem of vanishing-gradient. As a result, the feature extraction process is boosted up for its reuse, thereby reducing the number of parameters.

The CoroDet model by Hussain et al. [90] performed slightly better than our VGG16 model for binary classification between COVID-19 and normal. The authors achieved an accuracy of 99.1% in comparison to 97.24% by our model. However, our VGG16 model beat the CoroDet for three-class classification with an accuracy of 96.63% in comparison to 94.2%. Our model performed very close to the 97.97% of accuracy by the best performing Xception network for three-class classification applied by Jain et al. [91]. However, an advantage of our VGG16 network is that it is faster and takes less time for training. Furthermore, for the five-class classification, our VGG16 model outperformed other existing models with an accuracy of 92.70%.

### 7.3. A Special Note on Multiclass Frameworks for Pneumonia Classification

To date, most of the classification experiments for COVID-19 detection have been according to binary or three-class models. However, beyond COVID-19, a wide range of pneumonia exist among the population—including viral, bacterial, and tubercular. Therefore, it is vital to distinguish the COVID-19 from other diseases. A multiclass approach was apparently needed to classify COVID-19 from other pneumonia for the correct diagnosis of the patient. Our system is trained with the highest number of CXR images to date, includes most of the relevant pneumonia types, and is able to distinguish COVID-19 from other lung diseases with excellent accuracy.

### 7.4. Strengths, Weaknesses, and Extensions

The major strength of our system is its ability to detect COVID-19 very rapidly and it takes just a few seconds to provide results. Furthermore, the system is very cost-effective, as it requires only a patient’s chest X-ray scan which low-cost and readily available. Additionally, we have done six different types of classification experiments with consistently good accuracy that support our system’s robustness for practical applications.

One of the limitations of our system is its inability to detect the severity of the infection, partially due to collimator noise. One can adopt denoising methods [126] as part of preprocessing. Furthermore, predicting severity may help the physicians in treatment selection and thus in the fast and secure recovery of the patient. In addition, since we had a large database of CXR images, we did not perform k-fold cross-validation in which all images take part in training and testing at least once. In the extension of the work, we will make an effort for the advancement of the system, as it could be able to detect COVID-19 as well as the severity of the disease. In addition, we will include the heatmap images [127,128,129] of the disease, which will show the affected areas of the lungs. Broader advanced one-pass machine learning such as extreme learning machines [130] can be explored as more data are collected along with pruning methods [131,132,133] to lower the storage and improve the speed. This can also be extended for severity estimation [134] and application of an advanced image analysis solution such as stochastic imaging [135].

## 8. Conclusions

COVID-19 has become the foremost global challenge to save human life. Several healthcare organizations are struggling to discover effective solutions. However, artificial intelligence applications in computer-aided diagnosis (CAD) have proven their efficiency and importance in resolving several medical problems. Due to the presence of various types of pneumonia—such as viral, bacterial, tubercular, as well as COVID-19—a system was apparently needed for multiclass classification as current methods offer less reliable solutions. In this work, we have designed and applied seven highly efficient pre-trained convolutional neural networks—namely, VGG16, VGG19, DenseNet201, Xception, InceptionV3, NasnetMobile, and ResNet152—for classification of up to five classes by utilizing a large database of chest X-ray scans. For the first time ever, we have performed the binary classification into COVID-19 vs. bacterial pneumonia and COVID-19 vs. tuberculosis disease classes and achieved powerful accuracies of 99.84% by Densenet201 and 99.31% by VGG16 model, respectively. Our NasnetMobile and VGG16 models outperformed other existing methods for the binary (COVID-19 vs. viral pneumonia) and five-class (COVID-19, viral pneumonia, bacterial pneumonia, tuberculosis, and normal) classification with an accuracy of 99.80% and 92.70%, respectively. Performing with a remarkably high level of accuracy, the proposed models can provide an alternative to the current diagnostic methods for COVID-19 with a more accurate, cost-effective, and readily available system. The system may promisingly contribute to the fast diagnosis of patients, consequently lowering the medical load.

## Figures and Tables

**Figure 1 diagnostics-12-00652-f001:**
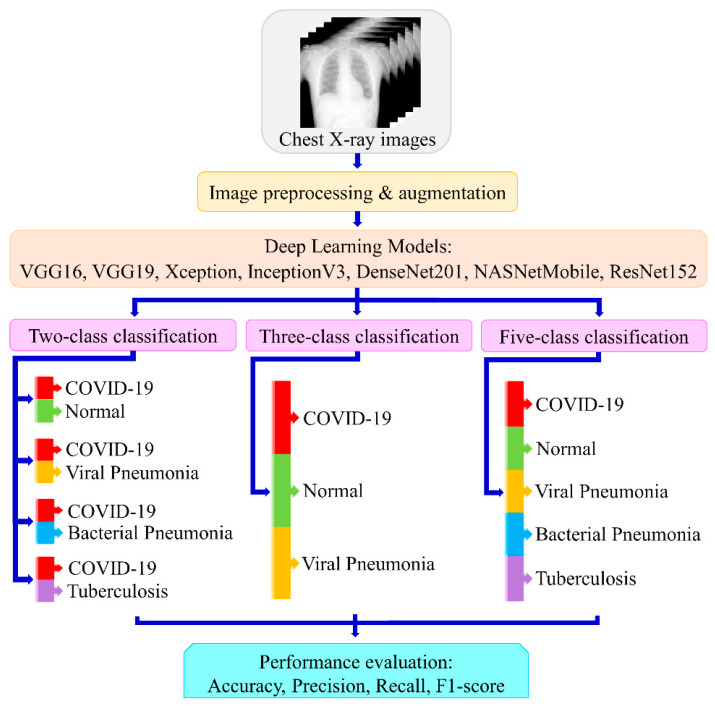
Overall schematic diagram of the proposed method for multiclass scenario using seven deep learning models.

**Figure 2 diagnostics-12-00652-f002:**
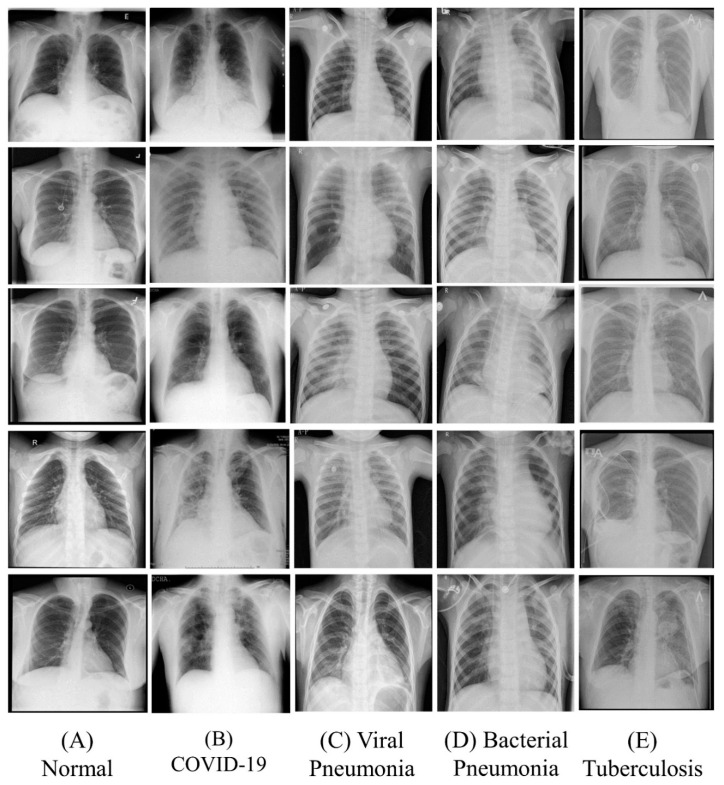
Sample chest X-ray images from each class.

**Figure 3 diagnostics-12-00652-f003:**
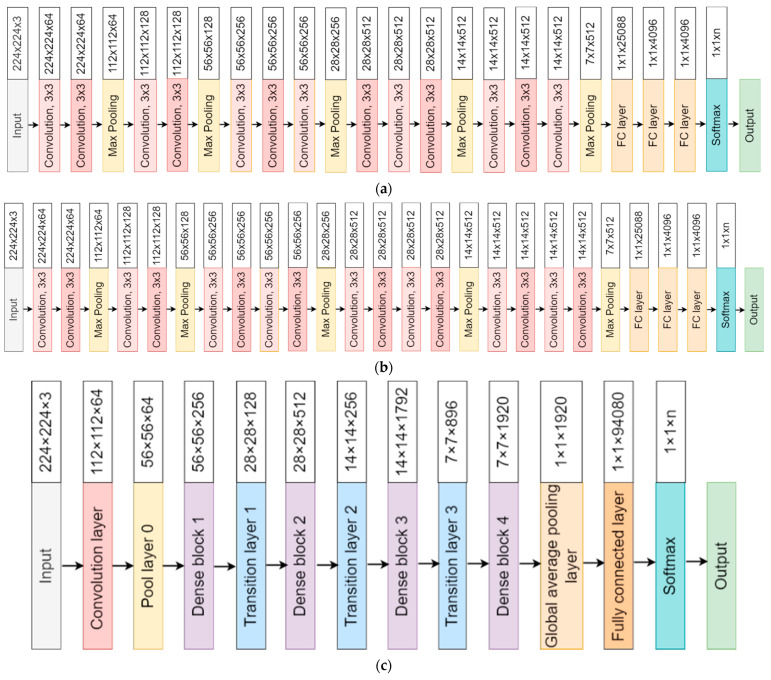
(**a**) VGG16 transfer learning architecture. (**b**) VGG19 transfer learning architecture. (**c**) DenseNet201 transfer learning architecture. (**d**) Xception transfer learning architecture. (**e**) InceptionV3 transfer learning architecture. (**f**) NasnetMobile transfer learning architecture. (**g**) ResNet152 transfer leaning architecture.

**Figure 4 diagnostics-12-00652-f004:**
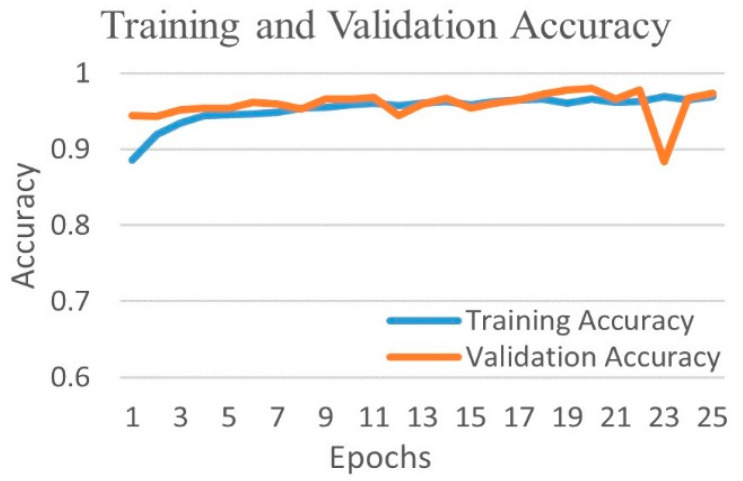
Training and validation accuracy of best performing VGG16 network for COVID-19 and normal class.

**Figure 5 diagnostics-12-00652-f005:**
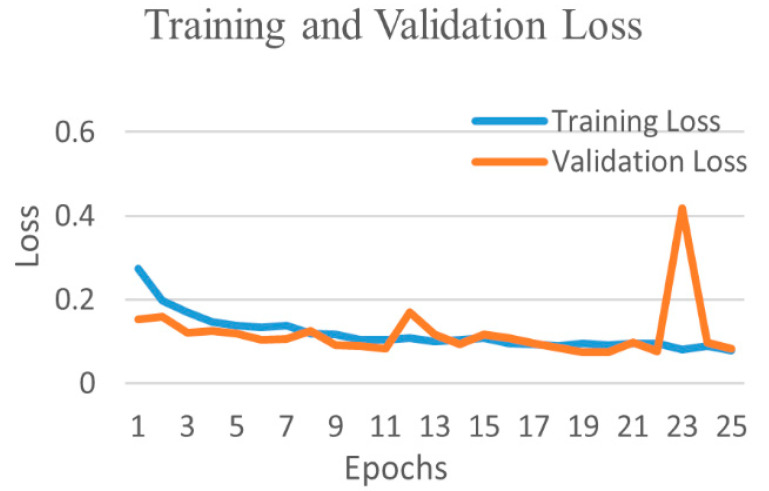
Training and validation loss of best performing VGG16 network for COVID-19 and normal class.

**Figure 6 diagnostics-12-00652-f006:**
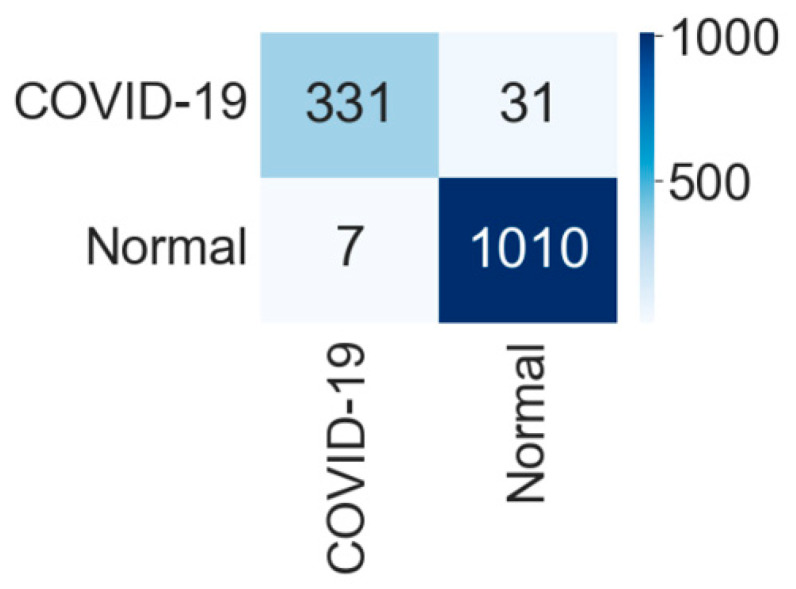
Confusion matrix for the classification into COVID-19 and normal by VGG16.

**Figure 7 diagnostics-12-00652-f007:**
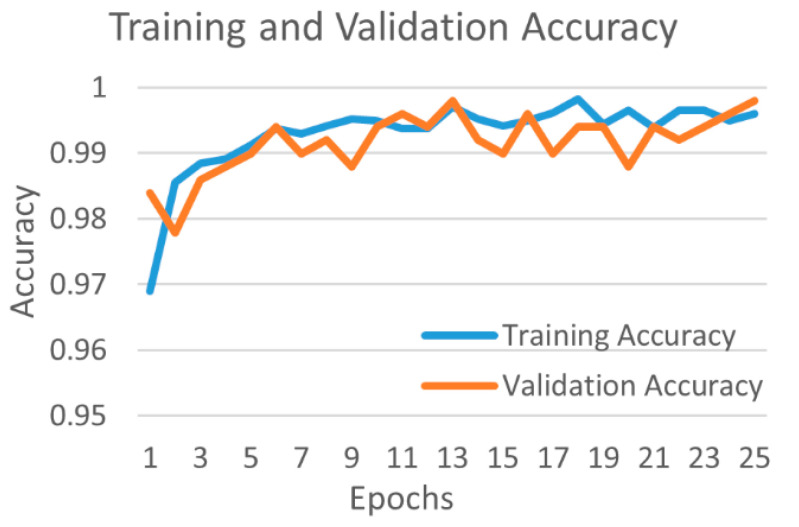
Training and validation accuracy of best performing NasnetMobile model for COVID-19 and viral pneumonia class.

**Figure 8 diagnostics-12-00652-f008:**
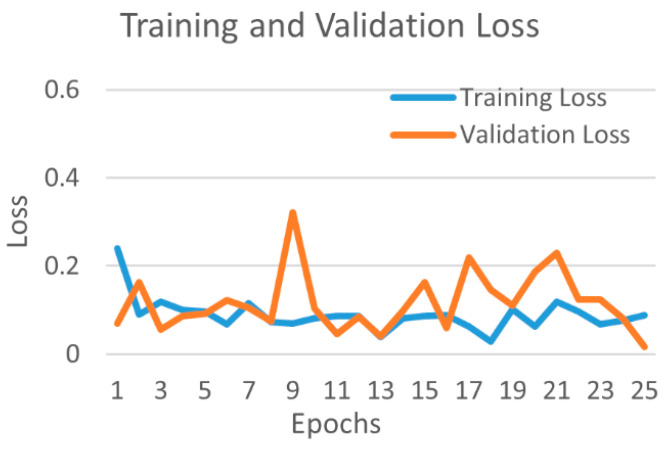
Training and validation loss of best performing NasnetMobile model for COVID-19 and viral pneumonia class.

**Figure 9 diagnostics-12-00652-f009:**
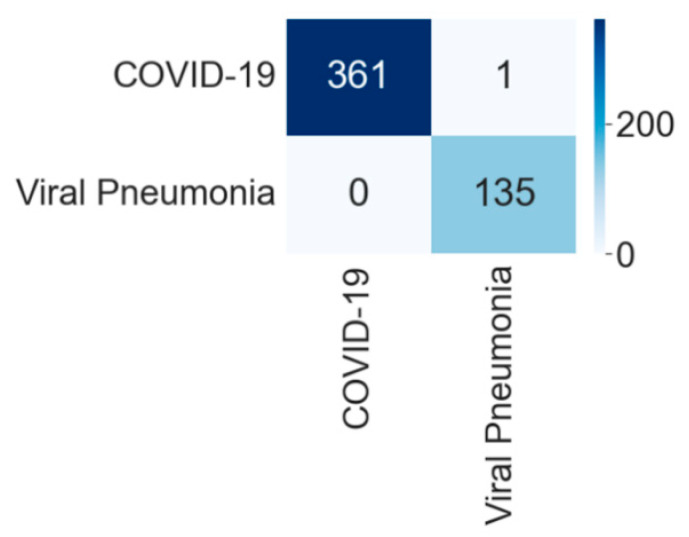
Confusion matrix for the classification into COVID-19 and viral pneumonia by NasnetMobile.

**Figure 10 diagnostics-12-00652-f010:**
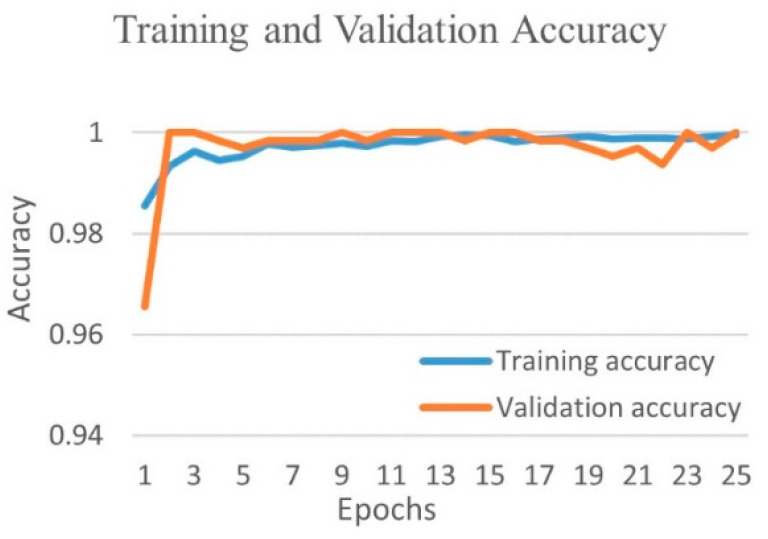
Training and validation accuracy of best performing DenseNet201 model for COVID-19 and bacterial pneumonia class.

**Figure 11 diagnostics-12-00652-f011:**
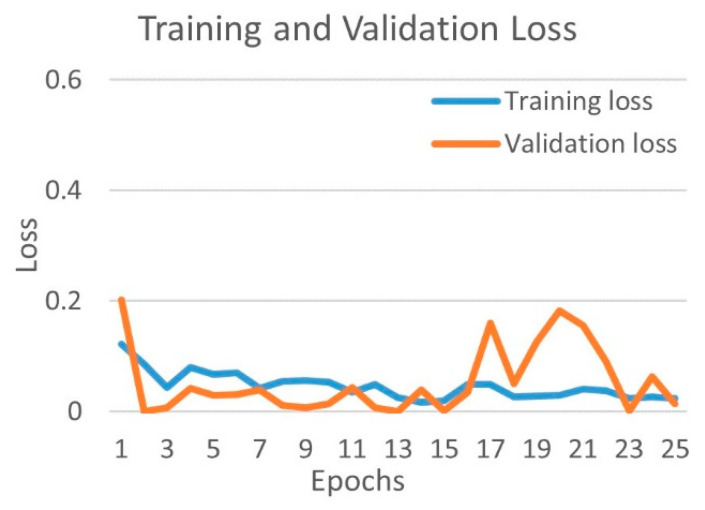
Training and validation loss of best performing DenseNet201 model for COVID-19 and bacterial pneumonia class.

**Figure 12 diagnostics-12-00652-f012:**
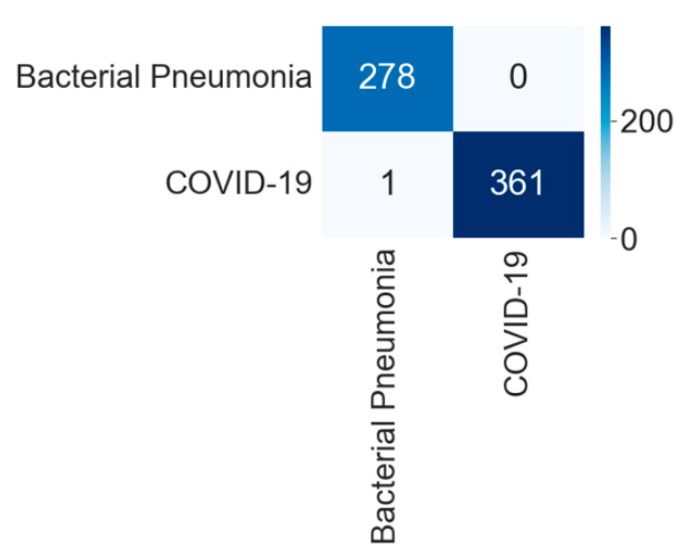
Confusion matrix for the classification into COVID-19 and bacterial pneumonia by DenseNet201.

**Figure 13 diagnostics-12-00652-f013:**
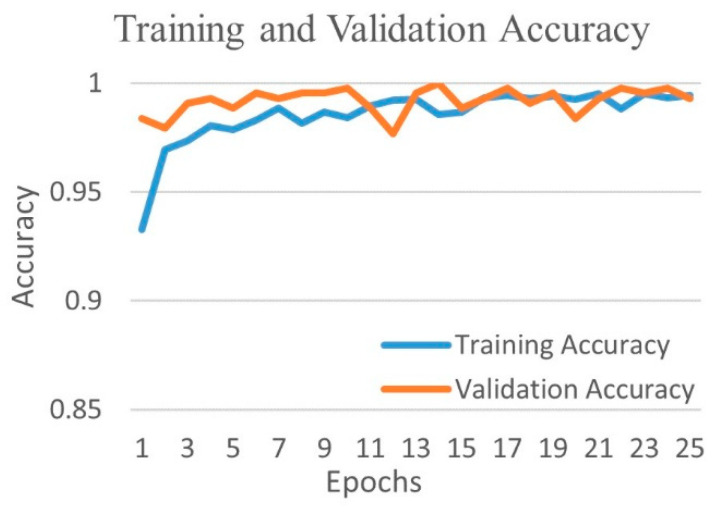
Training and validation accuracy of best performing VGG16 model for COVID-19 and tuberculosis class.

**Figure 14 diagnostics-12-00652-f014:**
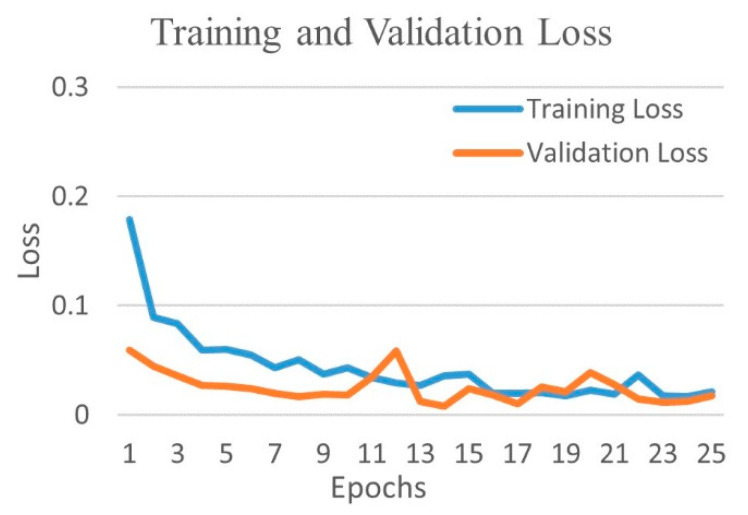
Training and validation loss of best performing VGG16 model for COVID-19 and tuberculosis class.

**Figure 15 diagnostics-12-00652-f015:**
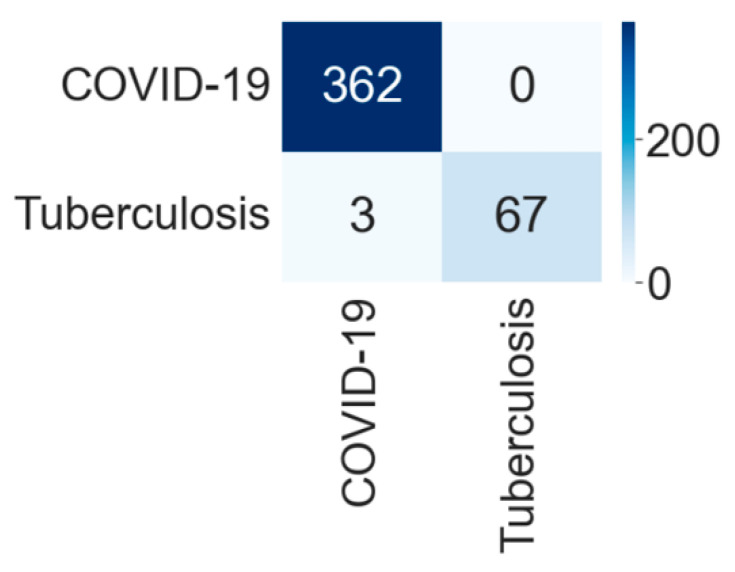
Confusion matrix for the classification into COVID-19 and tuberculosis by VGG16.

**Figure 16 diagnostics-12-00652-f016:**
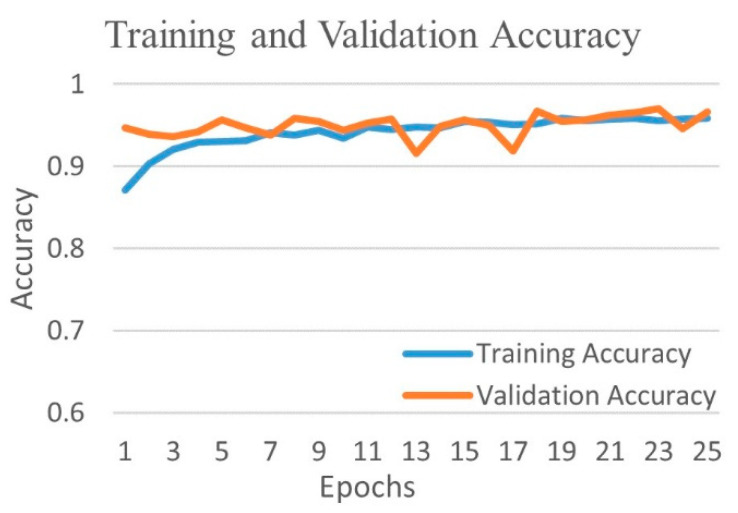
Training and validation accuracy of best performing VGG16 model for three-class experiment.

**Figure 17 diagnostics-12-00652-f017:**
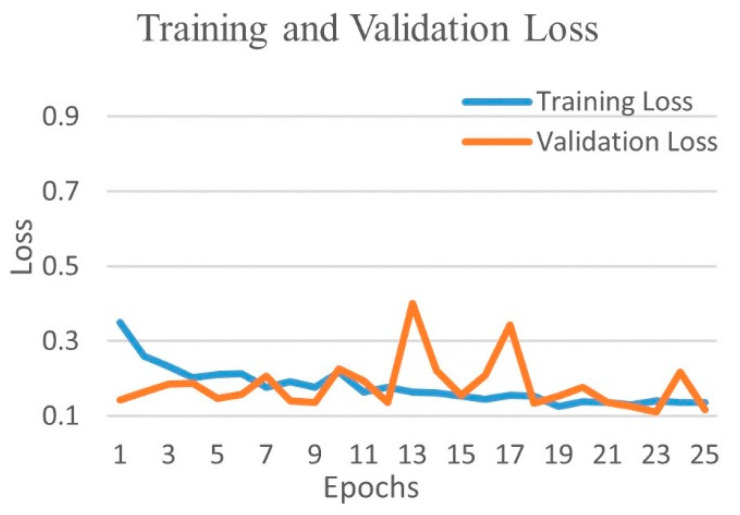
Training and validation loss of best performing VGG16 model for three-class experiment.

**Figure 18 diagnostics-12-00652-f018:**
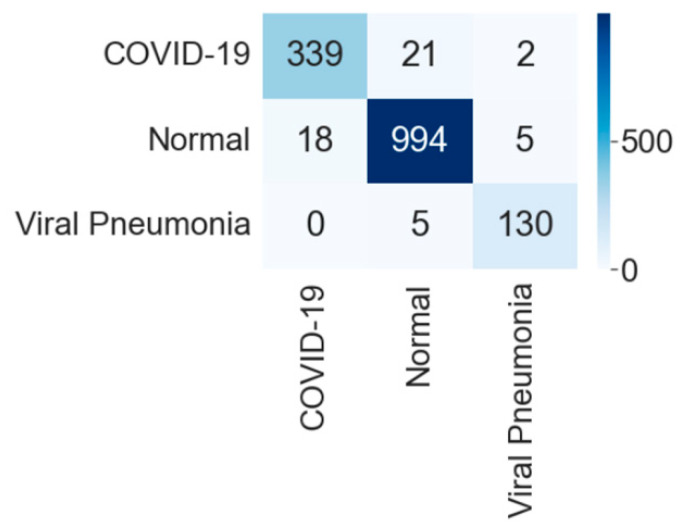
Confusion matrix for three-class classification by VGG16.

**Figure 19 diagnostics-12-00652-f019:**
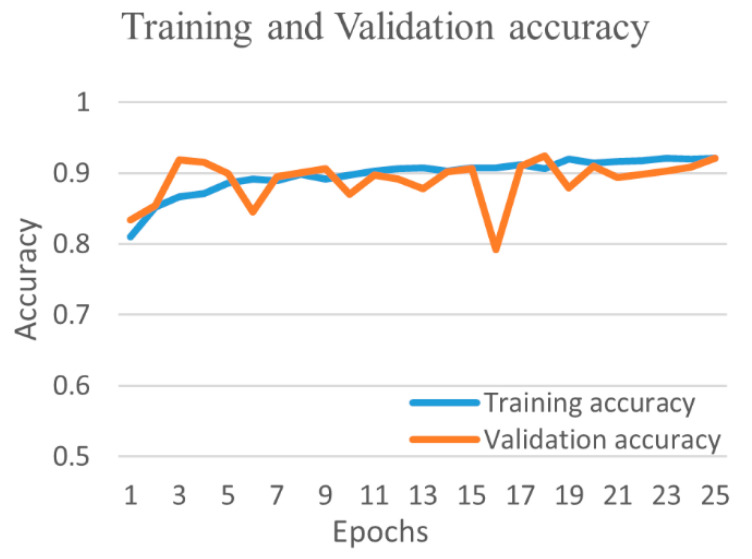
Training and validation accuracy of best performing VGG16 model for five-class.

**Figure 20 diagnostics-12-00652-f020:**
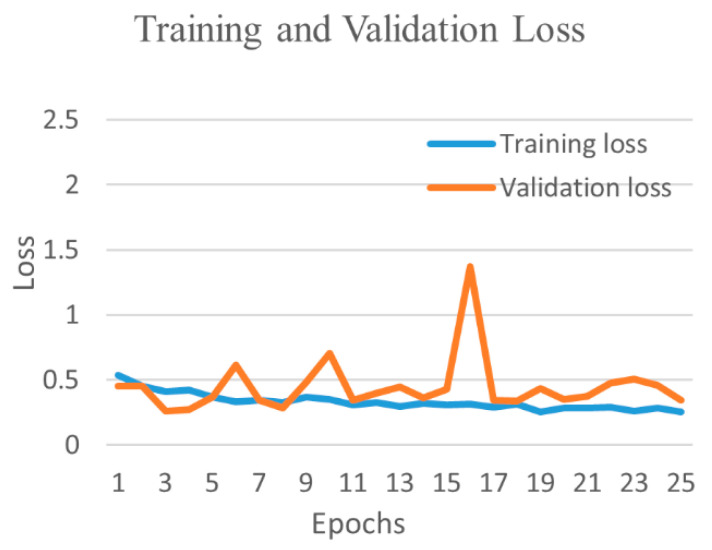
Training and validation loss of best performing VGG16 model for five-class.

**Figure 21 diagnostics-12-00652-f021:**
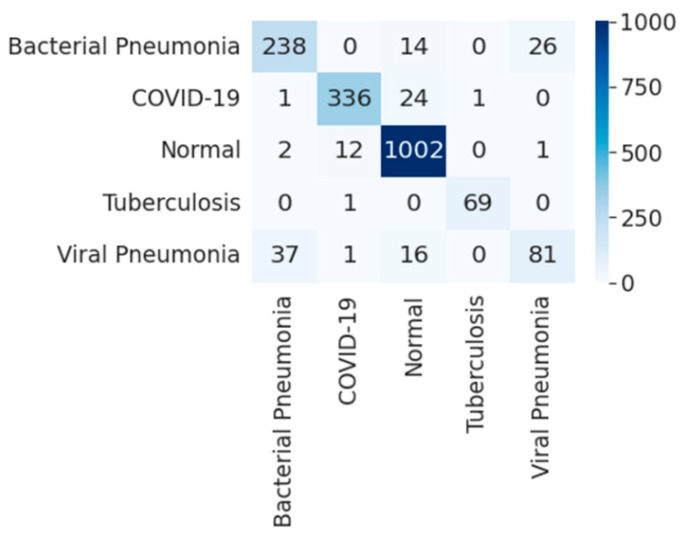
Confusion matrix for five-class classification by VGG16.

**Figure 22 diagnostics-12-00652-f022:**
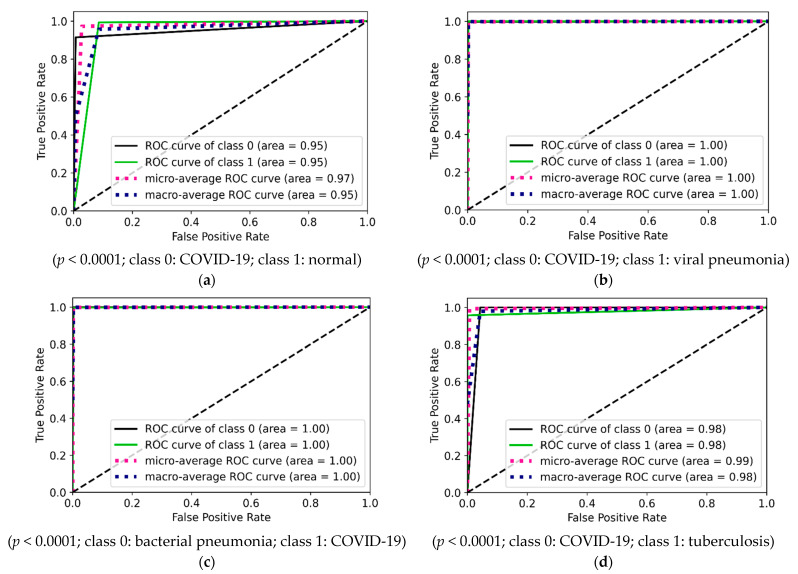
ROC curves and AUC values for two-class experiments: (**a**) COVID-19 and normal by VGG16; (**b**) COVID-19 and viral pneumonia by NasnetMobile; (**c**) COVID-19 and bacterial pneumonia by Densenet201; (**d**) COVID-19 and tuberculosis by VGG16.

**Figure 23 diagnostics-12-00652-f023:**
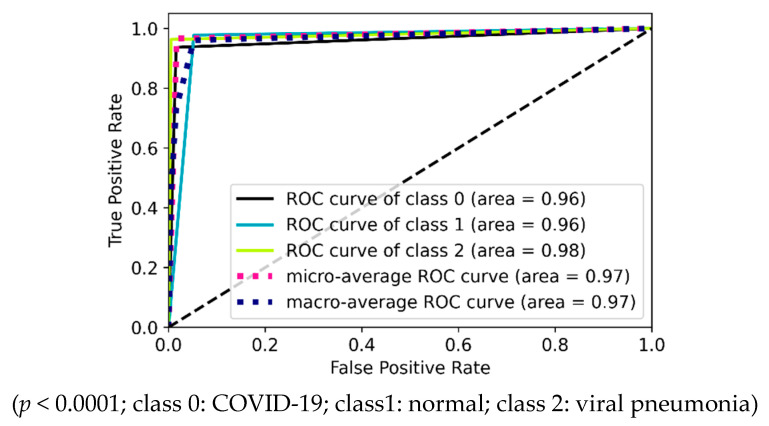
ROC curves and AUC values for three-class experiment by VGG16.

**Figure 24 diagnostics-12-00652-f024:**
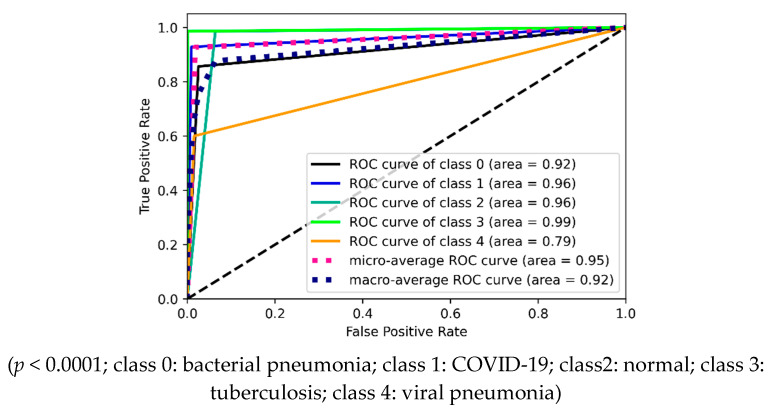
ROC curves and AUC values for five-class experiment by VGG16.

**Figure 25 diagnostics-12-00652-f025:**
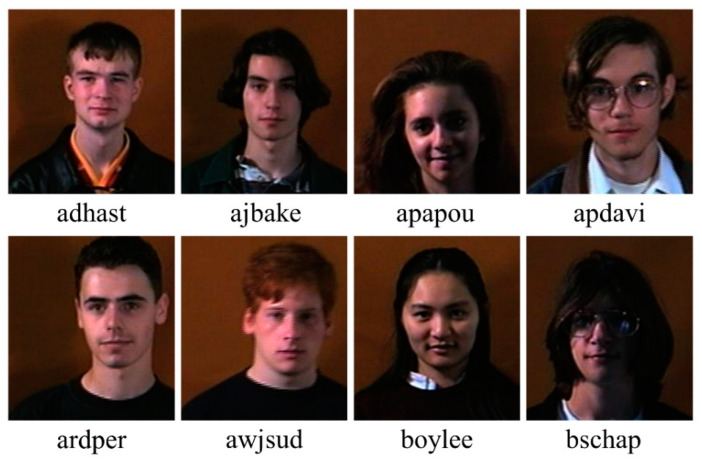
Sample images from the first eight classes of Faces95 database.

**Figure 26 diagnostics-12-00652-f026:**
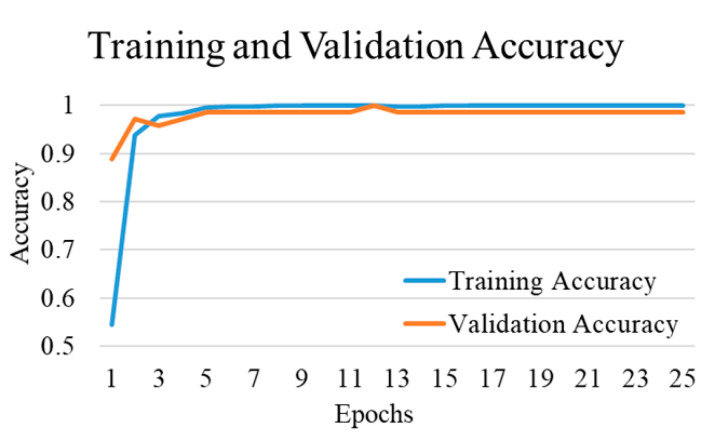
Training and validation accuracy of best performing VGG16 model for Faces95 images.

**Figure 27 diagnostics-12-00652-f027:**
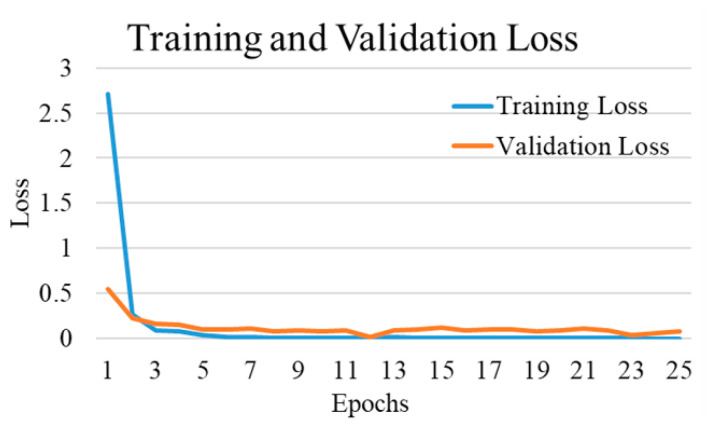
Training and validation loss of best performing VGG16 model for Faces95 images.

**Table 1 diagnostics-12-00652-t001:** Experimental steps and class-wise distribution of chest X-ray images.

Experimental Steps	Normal	COVID-19	Viral Pneumonia	Bacterial Pneumonia	Tuberculosis	Total
Training	8133	2887	1075	2224	560	14,879
Validation	1017	362	135	278	70	1862
Testing	1017	362	135	278	70	1862

**Table 2 diagnostics-12-00652-t002:** Weighted average of performance metrics by different deep learning models for COVID-19 and normal classification.

CNN Models	Accuracy	Precision	Recall	F1-Score
VGG16	97.24	97.26	97.24	97.21
VGG19	94.85	94.94	94.85	94.72
Xception	88.69	90.03	88.69	87.58
InceptionV3	93.33	93.32	93.33	93.32
DenseNet201	96.01	96.00	96.01	95.96
NasnetMobile	92.39	92.60	92.39	92.06
ResNet152	78.75	82.85	78.75	73.02

**Table 3 diagnostics-12-00652-t003:** Weighted average of performance metrics by different deep learning models for COVID-19 and viral pneumonia classification.

CNN Models	Accuracy	Precision	Recall	F1-Score
VGG16	99.60	99.60	99.60	99.60
VGG19	99.20	99.20	99.20	99.19
Xception	99.40	99.40	99.40	99.40
InceptionV3	98.99	99.01	98.99	99.00
Densenet201	99.40	99.40	99.40	99.40
NasnetMobile	99.80	99.80	99.80	99.80
Resnet152	97.79	97.80	97.79	97.77

**Table 4 diagnostics-12-00652-t004:** Weighted average of performance metrics by different deep learning models for COVID-19 and bacterial pneumonia classification.

CNN Models	Accuracy	Precision	Recall	F1-Score
VGG16	99.22	99.22	99.22	99.22
VGG19	98.75	98.76	98.75	98.75
Xception	99.06	99.08	99.06	99.06
InceptionV3	99.53	99.53	99.53	99.53
Densenet201	99.84	99.84	99.84	99.84
NasnetMobile	99.53	99.53	99.53	99.53
Resnet152	98.59	98.60	98.59	98.59

**Table 5 diagnostics-12-00652-t005:** Weighted average of performance metrics by different deep learning models for COVID-19 and tuberculosis classification.

CNN Models	Accuracy	Precision	Recall	F1-Score
VGG16	99.31	99.31	99.31	99.30
VGG19	99.07	99.07	99.07	99.07
Xception	99.07	99.07	99.07	99.07
InceptionV3	98.38	98.47	98.38	98.40
Densenet201	98.84	98.88	98.84	98.85
NasnetMobile	93.75	95.15	93.75	94.09
Resnet152	91.20	92.25	91.20	91.56

**Table 6 diagnostics-12-00652-t006:** Weighted average of performance metrics by different deep learning models for three-class classification into COVID-19, viral pneumonia, and normal.

CNN Models	Accuracy	Precision	Recall	F1-Score
VGG16	96.63	96.63	96.63	96.63
VGG19	91.94	92.49	91.94	91.63
Xception	91.68	91.64	91.68	91.54
InceptionV3	92.54	92.47	92.54	92.43
Densenet201	95.51	95.61	95.51	95.44
NasnetMobile	92.93	93.32	92.93	92.96
Resnet152	77.21	84.70	77.21	78.57

**Table 7 diagnostics-12-00652-t007:** Weighted average of performance metrics by different deep learning models for five-class classification into COVID-19, viral pneumonia, bacterial pneumonia, tuberculosis, and normal.

CNN Models	Accuracy	Precision	Recall	F1-Score
VGG16	92.70	92.41	92.70	92.47
VGG19	89.04	90.37	89.04	87.00
Xception	83.35	84.83	83.35	80.61
InceptionV3	84.00	85.54	84.00	83.44
Densenet201	89.10	89.80	89.10	88.42
NasnetMobile	87.76	88.05	87.76	86.65
Resnet152	74.70	76.80	74.70	71.60

**Table 8 diagnostics-12-00652-t008:** Weighted average of performance metrics by different deep learning networks for facial images classification.

CNN Models	Accuracy	Precision	Recall	F1-Score
VGG16	98.61	99.07	98.61	98.52
VGG19	96.53	97.45	96.53	96.34
Xception	93.06	93.75	93.06	92.18
InceptionV3	95.83	97.22	95.83	95.56
DenseNet201	96.53	97.69	96.53	96.30
NasnetMobile	93.06	95.60	93.06	92.82
ResNet152	75.69	76.50	75.69	80.13

**Table 9 diagnostics-12-00652-t009:** Benchmarking table showing state-of-the-art methods and comparing them against the proposed model.

Author and Year	Method and Models	Number of Images Used	Classification Accuracy				AUC ^1^
			Two-Class	Three-Class ^2^	Four-Class ^3^	Five-Class ^4^	
Nayak et al. (2020) [100]	Method: CNN with transfer learning Model: ResNet-34	C ^5^: 203 Total: 406	C ^5^ & N ^6^: 98.33%	NA ^7^	NA	NA	C & N: 0.98
Choudhury et al. (2020) [88]	Method: CNN with transfer learning Model: CheXNet	C: 423 Total: 3487	NA	97.74%	NA	NA	NA
Jain et al. (2020) [91]	Method: CNN with transfer learning Model: Xception	C: 490 Total: 6432	NA	97.97%	NA	NA	NA
Bhattacharyya et al. (2021) [101]	Method: ML ^8^ + DL ^9^ DL model: VGG-19 ML model: Random Forest	C: 342 Total: 1029	NA	96.6%	NA	NA	NA
Nikolaou et al. (2021) [103]	Method: CNN with transfer learning Model: EfficientNetB0	C: 3616 Total: 15,153	C & N: 95%	93%	NA	NA	NA
Yang et al. (2021) [99]	Method: CNN with transfer learning Model: VGG16	C: 3616 Total: 8461	C & N: 98% C & VP ^10^: 99%	97%	NA	NA	NA
Khan et al. (2020) [96]	Method: deep learning Model: Coronet (novel CNN)	C: 284 Total: 1251	NA	95%	89.6%	NA	NA
Hussain et al. (2020) [90]	Method: deep learning Model: CoroDet (novel CNN)	C: 500 Total: 2100	C & N: 99.1%	94.2%	91.2%	NA	NA
Oh et al. (2020) [104]	Method: CNN with transfer learning Model: ResNet-18	C: 180 Total: 502	NA	NA	88.9%	NA	NA
Timemy et al. (2021) [105]	Method: ML + DL DL model: ResNet-50 ML model: ESD ^11^	C: 435 Total: 2186	NA	NA	NA	91.6%	NA
Proposed work (Nillmani et al.)	Method: CNN with transfer learning Model: VGG16, NasnetMobile, DenseNet201	C: 3611 Total: 18,603	C & N: 97.24% ^12^ C & VP: 99.80% ^13^ C & BP ^14^: 99.84% ^15^ C & T ^16^: 99.31% ^12^	96.63% ^12^	NA	92.70% ^12^	C & N: 0.95 ^12^ C & VP: 1.0 ^13^ C & BP: 1.0 ^15^ C & T: 0.98 ^12^ Three-class ^2^: 0.97 ^12^ Five-class ^4^: 0.92 ^12^

^1^ Area under the ROC Curve; ^2^ COVID-19, viral pneumonia and normal; ^3^ COVID-19, viral pneumonia, bacterial pneumonia, and normal; ^4^ COVID-19, viral pneumonia, bacterial pneumonia, tuberculosis, and normal; ^5^ COVID-19; ^6^ Normal; ^7^ Not applicable as authors have not performed such type of experiment; ^8^ Machine learning; ^9^ Deep learning; ^10^ Viral pneumonia; ^11^ Ensemble subspace discriminant; ^12^ Acheived by VGG16; ^13^ Acheived by NasnetMobile; ^14^ bacterial pneumonia; ^15^ Acheived by DenseNet201; ^16^ tuberculosis.

## Data Availability

The dataset used in this study can be found in references [106,107,108].

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
