# Peer review of "Four Types of Multiclass Frameworks for Pneumonia Classification and Its Validation in X-ray Scans Using Seven Types of Deep Learning Artificial Intelligence Models"

_diagnostics, 2022, doi:10.3390/diagnostics12030652_

Round 1

Reviewer 1 Report

Review Report

Article ID: diagnostics-1628219

Article Title: Four Types of Multiclass Frameworks for Pneumonia Classification and its Validation in X-ray Scans using Seven Types of Deep Learning Artificial Intelligence Models

The paper proposes a deep learning AI-based automatic multiclass detection and classification of pneumonia from chest X-ray images that are available. The study applied seven pre-trained convolutional neural networks for classification up to five classes of pneumonia.

Although massive work has been done. However, the paper in the current format looks like a lab report as lots of information was added. At the publication level authors should be selective and precise in which I have the following suggestions:

1. Please correct all typos and grammatical mistakes in the paper and add missing captions such as Figure 10 and make sure that Figures numbers are correct.

  1. No references for Equations i to vi so please add references.
  2. Poor representation of results especially those generated on excel such as 4, 5, 7,8 .....etc; you can use a subplot and combine more than one graph together and make sure you provide sufficient description under each graph. 

Author Response

Response to Reviewers

Four Types of Multiclass Frameworks for Pneumonia Classification and its Validation in X-ray Scans using Seven Types of Deep Learning Artificial Intelligence Models

Manuscript ID: diagnostics- 1628219

 Authors: Nillmani et al.

Date: 04 March 2022

Reviewer 1

  1. Please correct all typos and grammatical mistakes in the paper and add missing captions such as Figure 10 and make sure that Figures numbers are correct.

Authors: Thank you for your valuable feedback and for pointing out the error. We have now corrected the grammatical mistakes in the entire manuscript.

Further, we have also added the missing caption in Figure 10 (please see page no.16 of the revised manuscript).  Finally, we also have checked each and every Figure and Table and their numbers with captions again.

Thank you once again for improving the readability of the manuscript.

  1. No references for Equations i to vi so please add references.

Authors: We have now called the above equations at the proper places in the paragraph. Please see page no. 12 in the revised manuscript. Further, we have also quoted the references for the corresponding equations (i-vi).

Thank you again very much.

  1. Poor representation of results especially those generated on excel such as 4, 5, 7,8 .....etc; you can use a subplot and combine more than one graph together and make sure you provide sufficient description under each graph.

Authors: Thank you again for your valuable feedback.

We have now updated these figures and combined them. Figures 4 and 5 have been combined into one figure (see Figure A below) as suggested by this reviewer. Similarly, Figures 7 and 8 have been combined into one figure, accordingly. Note that, since after combining the accuracy curves nearly overlap, thus, we have given a zoomed version of the accuracy curve in the appendix (see Figure B below).

Thank you very much for your support in improving the readability of the manuscript.

                         Figure A. Combined accuracy and loss function for training and validation protocols.

Figure B. Zoomed version of the training and validation accuracy (blue-training accuracy and orange-validation accuracy).

Thank you

The pdf with the images has been attached.

Reviewer 2 Report

Four Types of Multiclass Frameworks for Pneumonia Classification and its Validation in X-ray Scans using Seven Types of Deep Learning Artificial Intelligence Models

Manuscript ID: diagnostics-1628219

The main objective of this study is to propose a deep learning AI-based automatic multiclass detection and classification of pneumonia from chest X-ray images that are readily available and highly cost-effective. The study has designed  and  applied  seven  highly  efficient  pre-trained  convolutional  neural  networks  namely,  VGG16, VGG19,  DenseNet201,  Xception,  InceptionV3,  NasnetMobile,  and  ResNet152  for  classification  up  to  five classes  of  pneumonia.

The approach of the study appears very original, and the contents of the manuscript is quite interesting by his methodology.

The manuscript reads smoothly and is easy to understand. The aims, scope, and results of the study are clearly stated. I have very much enjoyed reading this paper. I find it interesting and clearly written and satisfying also all the other publication criteria of the “Journal Diagnostics”. The study provides a very valuable addition to this line of research, and adds relevantly to the subject with additional original findings. I thus find that this paper definitively delivers results that will surely be of interest to the readership of the “Journal Diagnostics”. I recommend the publication of this paper after minor revision: Need more details about the validation of the model. The authors need to explain this part by a chapter and to show the limits of this kind of approach. I recommend the addition of the following references that in my opinion are missed and can help the discussion section.

*Melo-Oliveira, M.E., Sá-Caputo, D. et al. 2021. Reported quality of life in countries with cases of COVID19: a systematic review. Expert Review of Respiratory Medicine, 15(2), pp. 213-220

Davahli MR, Karwowski W, Fiok K, Murata A et al. The COVID-19 Infection Diffusion in the US and Japan: A Graph-Theoretical Approach. Biology. 2022; 11(1):125. https://doi.org/10.3390/biology11010125

*N. Sapkota et al., 2021. "The Chaotic Behavior of the Spread of Infection During the COVID-19 Pandemic in the United States and Globally," in IEEE Access, vol. 9, pp. 80692-80702, 2021, doi: 10.1109/ACCESS.2021.3085240. 

Author Response

Response to Reviewers

Four Types of Multiclass Frameworks for Pneumonia Classification and its Validation in X-ray Scans using Seven Types of Deep Learning Artificial Intelligence Models

Manuscript ID: diagnostics- 1628219

 Authors: Nillmani et al.

Date: 04 March 2022

Reviewer 2

Need more details about the validation of the model. The authors need to explain this part by a chapter and to show the limits of this kind of approach. I recommend the addition of the following references that in my opinion are missed and can help the discussion section.

*Melo-Oliveira, M.E., Sá-Caputo, D. et al. 2021. Reported quality of life in countries with cases of COVID19: a systematic review. Expert Review of Respiratory Medicine, 15(2), pp. 213-220

Davahli MR, Karwowski W, Fiok K, Murata A et al. The COVID-19 Infection Diffusion in the US and Japan: A Graph-Theoretical Approach. Biology. 2022; 11(1):125. https://doi.org/10.3390/biology11010125

*N. Sapkota et al., 2021. "The Chaotic Behavior of the Spread of Infection During the COVID-19 Pandemic in the United States and Globally," in IEEE Access, vol. 9, pp. 80692-80702, 2021, doi: 10.1109/ACCESS.2021.3085240. 

Author: Thank you so much for your valuable feedback. We have divided the answer in two parts: Part A discuss about the validation while Part B discusses about the inclusion of the three references.

Part A: About Validation Protocol

Validation is an import component of the AI and thus, we have first tested our model on the 10% of the unseen dataset that were not involved in any of the training protocols.

Next, we have implemented six different types of classification experiments with the different number of classes and achieved consistently a significant test accuracy. Nevertheless, finally we have done a separate section naming “scientific validation” where we have tested each model on a different dataset i.e., facial biometric images. Our models depicted encouraging results meeting the requirements, demonstrating highly accurate results on the Faces95 dataset. We have also added a subsection 6.4 called strength, weaknesses and extension, where we have discussed about the limitations of our AI approach.

Part B: Inclusion of the three references

We have also examined these three references recommended by this reviewer. We have now added a new paragraph in the beginning of the discussion section (page 26 of the revised manuscript) elaborating an insight about these three references.

In lieu of the above discussions, this paragraph reads as follows:

“The breakdown of coronaviruses and their fast spread has put the world in a very tough situation. The quality of life due to COVID-19 has gotten worse, including increased anxiety and depression [119]. The chaotic behaviour of COVID-19 has caused it to spread the virus as a nonlinear infection throughout different parts of the world [120]. Certain places of the world have a larger number of infections, while some have low. Further, some have greater intensity and severity, while at some places it has smaller intensity and severity. Additionally, on the cases related to a daily basis, studies have shown no significant correlation in the diffusion of disease in the different parts of the world [121]. Thus, it makes it difficult to predict, prepare, and combat the outbreak of the disease variants. To find the solutions and overcome the limitations such as unavailability of RT-PCR tests, delayed results, and its high costs, our strategy of deep learning-based transfer models for the classification of chest X-ray images to detect COVID-19 is proving to be the most effective.”

Thank you very much for your support in improving the readability of the manuscript.

[119] Melo-Oliveira, M.E., Sá-Caputo, D. et al. 2021. Reported quality of life in countries with cases of COVID19: a systematic review. Expert Review of Respiratory Medicine, 15(2), pp. 213-220.

[120] N. Sapkota et al., 2021. "The Chaotic Behavior of the Spread of Infection During the COVID-19 Pandemic in the United States and Globally," in IEEE Access, vol. 9, pp. 80692-80702, 2021, doi: 10.1109/ACCESS.2021.3085240. 

[121] Davahli MR, Karwowski W, Fiok K, Murata A et al. The COVID-19 Infection Diffusion in the US and Japan: A Graph-Theoretical Approach. Biology. 2022; 11(1):125. https://doi.org/10.3390/biology11010125.

Thank you.
